# Monotone operator equilibrium networks

**Ezra Winston**
School of Computer Science
Carnegie Mellon University
Pittsburgh, United States
ewinston@cs.cmu.edu

**J. Zico Kolter**
School of Computer Science
Carnegie Mellon University
& Bosch Center for AI
Pittsburgh, United States
zkolter@cs.cmu.edu

## Abstract

Implicit-depth models such as Deep Equilibrium Networks have recently been shown to match or exceed the performance of traditional deep networks while being much more memory efficient. However, these models suffer from unstable convergence to a solution and lack guarantees that a solution exists. On the other hand, Neural ODEs, another class of implicit-depth models, do guarantee existence of a unique solution but perform poorly compared with traditional networks. In this paper, we develop a new class of implicit-depth model based on the theory of monotone operators, the Monotone Operator Equilibrium Network (monDEQ). We show the close connection between finding the equilibrium point of an implicit network and solving a form of monotone operator splitting problem, which admits efficient solvers with guaranteed, stable convergence. We then develop a parameterization of the network which ensures that all operators remain monotone, which guarantees the existence of a unique equilibrium point. Finally, we show how to instantiate several versions of these models, and implement the resulting iterative solvers, for structured linear operators such as multi-scale convolutions. The resulting models vastly outperform the Neural ODE-based models while also being more computationally efficient. Code is available at http://github.com/locuslab/monotone_op_net.

## 1 Introduction

Recent work in deep learning has demonstrated the power of *implicit-depth* networks, models where features are created not by explicitly iterating some number of nonlinear layers, but by finding a solution to some implicitly defined equation. Instances of such models include the Neural ODE [8], which computes hidden layers as the solution to a continuous-time dynamical system, and the Deep Equilibrium (DEQ) Model [5], which finds a fixed point of a nonlinear dynamical system corresponding to an effectively infinite-depth weight-tied network. These models, which trace back to some of the original work on recurrent backpropagation [2, 23], have recently regained attention since they have been shown to match or even exceed to performance of traditional deep networks in domains such as sequence modeling [5]. At the same time, these models show drastically improved memory efficiency over traditional networks since backpropagation is typically done analytically using the implicit function theorem, without needing to store the intermediate hidden layers.

However, implict-depth models that perform well require extensive tuning in order to achieve stable convergence to a solution. Obtaining convergence in DEQs requires careful initialization and regularization, which has proven difficult in practice [21]. Moreover, solutions to these models are not guaranteed to exist or be unique, making the output of the models potentially ill-defined. While Neural ODEs [8] do guarantee existence of a unique solution, training remains unstable since the ODE problems can become severely ill-posed [10]. Augmented Neural ODEs [10] improve the stability of Neural ODEs by learning ODEs with simpler flows, but neither model achieves efficient

convergence nor performs well on standard benchmarks. Crucial questions remain about how models can have guaranteed, unique solutions, and what algorithms are most efficient at finding them.

In this paper, we present a new class of implicit-depth equilibrium model, the Monotone Operator Equilibrium Network (monDEQ), which guarantees stable convergence to a unique fixed point.[1] The model is based upon the theory of monotone operators [6, 26], and illustrates a close connection between simple fixed-point iteration in weight-tied networks and the solution to a particular form of monotone operator splitting problem. Using this connection, this paper lays the theoretical and practical foundations for such networks. We show how to parameterize networks in a manner that ensures all operators remain monotone, which establishes the existence and uniqueness of the equilibrium point. We show how to backpropagate through such networks using the implicit function theorem; this leads to a corresponding (linear) operator splitting problem for the backward pass, which also is guaranteed to have a unique solution. We then adapt traditional operator splitting methods, such as forward-backward splitting or Peaceman-Rachford splitting, to naturally derive algorithms for efficiently computing these equilibrium points.

Finally, we demonstrate how to practically implement such models and operator splitting methods, in the cases of typical feedforward, fully convolutional, and multi-scale convolutional networks. For convolutional networks, the most efficient fixed-point solution methods require an inversion of the associated linear operator, and we illustrate how to achieve this using the fast Fourier transform. The resulting networks show strong performance on several benchmark tasks, vastly improving upon the accuracy and efficiency of Neural ODEs-based models, the other implicit-depth models where solutions are guaranteed to exist and be unique.

## 2   Related work

**Implicit models in deep learning**   There has been a growing interest in recent years in implicit layers in deep learning. Instead of specifying the explicit computation to perform, a layer specifies some *condition* that should hold at the solution to the layer, such as a nonlinear equality, or a differential equation solution. Using the implicit function theorem allows for backpropagating through the layer solutions *analytically*, making these layers very memory efficient, as they do not need to maintain intermediate iterations of the solution procedure. Recent examples include layers that compute inference in graphical models [15], solve optimization problems [12, 3, 13, 1], execute model-based control policies [4], solve two-player games [20], solve gradient-based optimization for meta-learning [24], and many others.

**Stability of fixed-point models**   The issue of model stability has in fact been at the heart of much work in fixed-point models. The original work on attractor-style recurrent models, trained via recurrent backpropagation [2, 23], precisely attempted to ensure that the forward iteration procedure was stable. And indeed, much of the work in recurrent architectures such as LSTMs has focused on these issues of stability [14]. Recent work has revisited recurrent backpropagation in a similar manner to DEQs, with the similar aim of speeding up the computation of fixed points [19]. And other work has looked at the stability of implicit models [11], with an emphasis on guaranteeing the existence of fixed points, but focused on alternative stability conditions, and considered only relatively small-scale experiments. Other recent work has looked to use control-theoretic methods to ensure the stability of implicit models, [25], though again they consider only small-scale evaluations.

**Monotone operators in deep learning**   Although most work in the field of monotone operators is concerned with general convex analysis, the recent work of [9] does also highlight connections between deep networks and monotone operator problems. Unlike our current work, however, that work focused largely on the fact that many common non-linearities can be expressed via proximal operators, and analyzed traditional networks under the assumptions that certain of the operators were monotone, but did not address conditions for the networks to be monotone or algorithms for solving or backpropagating through the networks.

## 3   A monotone operator view of fixed-point networks

This section lays out our main methodological and theoretical contribution, a class of equilibrium networks based upon monotone operators. We begin with some preliminaries, then highlight the

basic connection between the fixed point of an "infinite-depth" network and an associated operator splitting problem; next, we propose a parameterization that guarantees the associated operators to be maximal monotone; finally, we show how to use operator splitting methods to both compute the fixed point and backpropagate through the fixed point efficiently.

## 3.1 Preliminaries

**Monotone operator theory** The theory of monotone operators plays a foundational role in convex analysis and optimization. Monotone operators are a natural generalization of monotone functions, which can be used to assess the convergence properties of many forms of iterative fixed-point algorithms. We emphasize that the majority of the work in this paper relies on well-known properties about monotone operators, and we refer to standard references on the topic including [6] and a less formal survey by [26]; we do include a brief recap of the definitions and results we require in Appendix A. Formally, an operator is a subset of the space $F \subseteq \mathbb{R}^n \times \mathbb{R}^n$; in our setting this will usually correspond to set-valued or single-valued function. Operator splitting approaches refer to methods for finding a zero in a sum of operators, i.e., find $x$ such that $0 \in (F + G)(x)$. There are many such methods, but the two we will use mainly in this work are *forward-backward* splitting (eqn. A9 in the Appendix) and *Peaceman-Rachford* splitting (eqn. A10). As we will see, both finding a network equilibrium point and backpropagating through it can be formulated as operator splitting problems, and different operator splitting methods will lead to different approaches in their application to our subsequent implicit networks.

**Deep equilibrium models** The monDEQ architecture is closely relate to the DEQ model, which parameterizes a "weight-tied, input-injected" network of the form $z_{i+1} = g(z_i, x)$, where $x$ denotes the input to the network, injected at each layer; $z_i$ denotes the hidden layer at depth $i$; and $g$ denotes a nonlinear function which is the same for each layer (hence the network is weight-tied). The key aspect of the DEQ model is that in this weight-tied setting, instead of forward iteration, we can simply use any root-finding approach to find an equilibrium point of such an iteration $z^* = g(z^*, x)$. Assuming the model is stable, this equilibrium point corresponds to an "infinite-depth fixed point" of the layer. The monDEQ architecture can be viewed as an instance of a DEQ model, but one that relies on the theory of monotone operators, and a specific paramterization of the network weights, to guarantee the existence of a unique fixed point for the network. Crucially, however, as is the case for DEQs, naive forward iteration of this model is *not* necessarily stable; we therefore employ operator splitting methods to develop provably (linearly) convergent methods for finding such fixed points.

## 3.2 Fixed-point networks as operator splitting

As a starting point of our analysis, consider the weight-tied, input-injected network in which $x \in \mathbb{R}^d$ denotes the input, and $z^k \in \mathbb{R}^n$ denotes the hidden units at layer $k$, given by the iteration[2]

$$z^{k+1} = \sigma(Wz^k + Ux + b) \tag{1}$$

where $\sigma : \mathbb{R} \to \mathbb{R}$ is a nonlinearity applied elementwise, $W \in \mathbb{R}^{n \times n}$ are the hidden unit weights, $U \in \mathbb{R}^{n \times x}$ are the input-injection weights and $b \in \mathbb{R}^n$ is a bias term. An equilibrium point, or fixed point, of this system is some point $z^\star$ which remains constant after an update:

$$z^\star = \sigma(Wz^\star + Ux + b). \tag{2}$$

We begin by observing that it is possible to characterize this equilibrium point exactly as the solution to a certain operator splitting problem, under certain choices of operators and activation $\sigma$. This can be formalized in the following theorem, which we prove in Appendix B:

**Theorem 1.** *Finding a fixed point of the iteration (1) is equivalent to finding a zero of the operator splitting problem $0 \in (F + G)(z^\star)$ with the operators*

$$F(z) = (I - W)(z) - (Ux + b), \quad G = \partial f \tag{3}$$

*and $\sigma(\cdot) = \mathrm{prox}_f^1(\cdot)$ for some convex closed proper (CCP) function $f$, where $\mathrm{prox}_f^\alpha$ denotes the proximal operator*

$$\mathrm{prox}_f^\alpha(x) \equiv \underset{z}{\mathrm{argmin}} \frac{1}{2}\|x - z\|_2^2 + \alpha f(z). \tag{4}$$

It is also well-established that many common nonlinearities used in deep networks can be represented as proximal operators of CCP functions [7, 9]. For example, the ReLU nonlinearity $\sigma(x) = [x]_+$ is the proximal operator of the indicator of the positive orthant $f(x) = I\{x \geq 0\}$, and tanh, sigmoid, and softplus all have close correspondence with proximal operators of simple expressions [7].

In fact, this method establishes that some seemingly unstable iterations can actually still lead to convergent algorithms. ReLU activations, for instance, have traditionally been avoided in iterative models such as recurrent networks, due to exploding or vanishing gradient problems and nonsmoothness. Yet this iteration shows that (with input injection and the above constraint on $W$), ReLU operators are perfectly well-suited to these fixed-point iterations.

### 3.3    Enforcing existence of a unique solution

The above connection is straightforward, but also carries interesting implications for deep learning. Specifically, we can establish the existence and uniqueness of the equilibirum point $z^\star$ via the simple sufficient criterion that $I - W$ is strongly monotone, or in other words[3] $I - W \succeq mI$ for some $m > 0$ (see Appendix A). The constraint is by no means a trivial condition. Although many layers obey this condition under typical initialization schemes, during training it is normal for $W$ to move outside this regime. Thus, the first step of the monDEQ architecture is to parameterize $W$ in such a way that it always satisfies this strong monotonicity constraint.

**Proposition 1.** *We have $I - W \succeq mI$ if and only if there exist $A, B \in \mathbb{R}^{n \times n}$ such that*

$$W = (1 - m)I - A^T A + B - B^T. \tag{5}$$

We therefore propose to simply parameterize $W$ directly in this form, by defining the $A$ and $B$ matrices directly. While this is an overparameterized form for a dense matrix, we could avoid this issue by, e.g. constraining $A$ to be lower triangular (making it the Cholesky factor of $A^T A$), and by making $B$ strictly upper triangular; in practice, however, simply using general $A$ and $B$ matrices has little impact upon the performance of the method. The parameterization does notably raise additional complications when dealing with convolutional layers, but we defer this discussion to Section 4.2.

### 3.4    Computing the network fixed point

Given the monDEQ formulation, the first natural question to ask is: how should we compute the equilibrium point $z^\star = \sigma(Wz^\star + Ux + b)$? Crucially, it can be the case that the simple forward iteration of the network equation (1) does *not* converge, i.e., the iteration may be unstable. Fortunately, monotone operator splitting leads to a number of iterative methods for finding these fixed points, which are guaranteed to converge under proper conditions. For example, the forward-backward iteration applied to the monotone operator formulation from Theorem 1 results exactly in a damped version of the forward iteration

$$z^{k+1} = \text{prox}_f^\alpha(z^k - \alpha((I - W)z^k - (Ux + b))) = \text{prox}_f^\alpha((1 - \alpha)z^k + \alpha(Wz^k + Ux + b)). \tag{6}$$

This iteration is guaranteed to converge linearly to the fixed point $z^\star$ provided that $\alpha \leq 2m/L^2$, when the operator $I - W$ is Lipschitz and strongly monotone with parameters $L$ (which is simply the operator norm $\|I - W\|_2$) and $m$ [26].

A key advantage of the monDEQ formulation is the flexibility to employ alternative operator splitting methods that converge much more quickly to the equilibrium. One such example is Peaceman-Rachford splitting which, when applied to the formulation from Theorem 1, takes the form

$$
\begin{aligned}
u^{k+1/2} &= 2z^k - u^k \\
z^{k+1/2} &= (I + \alpha(I - W))^{-1}(u^{k+1/2} - \alpha(Ux + b)) \\
u^{k+1} &= 2z^{k+1/2} - u^{k+1/2} \\
z^{k+1} &= \text{prox}_f^\alpha(u^{k+1})
\end{aligned}
\tag{7}
$$

where we use the explicit form of the resolvents for the two monotone operators of the model. The advantage of Peaceman-Rachford splitting over forward-backward is two-fold: 1) it typically converges in fewer iterations, which is a key bottleneck for many implicit models; and 2) it converges

| **Algorithm 1** Forward-backward equilibrium solving | **Algorithm 2** Peaceman-Rachford equilibrium solving |
|---|---|
| $z := 0; \quad \text{err} := 1$ | $z, u := 0; \quad \text{err} := 1; \quad V := (I + \alpha(I - W))^{-1}$ |
| **while** err $> \epsilon$ **do** | **while** err $> \epsilon$ **do** |
| $\quad z^+ := (1 - \alpha)z + \alpha(Wz + Ux + b)$ | $\quad u^{1/2} := 2z - u$ |
| $\quad z^+ := \text{prox}_f^\alpha(z^+)$ | $\quad z^{1/2} := V(u^{1/2} + \alpha(Ux + b))$ |
| $\quad \text{err} := \frac{\|z^+ - z\|_2}{\|z^+\|_2}$ | $\quad u^+ := 2z^{1/2} - u^{1/2}$ |
| $\quad z := z^+$ | $\quad z^+ := \text{prox}_f^\alpha(u^+)$ |
| **return** $z$ | $\quad \text{err} := \frac{\|z^+ - z\|_2}{\|z^+\|_2}$ |
| | $\quad z, u := z^+, u^+$ |
| | **return** $z$ |

for any $\alpha > 0$ [26], unlike forward-backward splitting which is dependent on the Lipschitz constant of $I - W$. The disadvantage of Peaceman-Rachford splitting, however, is that it requires an inverse involving the weight matrix $W$. It is not immediately clear how to apply such methods if the $W$ matrix involves convolutions or multi-layer models; we discuss these points in Section 4.2. A summary of these methods for computation of the forward equilibrium point is given in Algorithms 1 and 2.

### 3.5 Backprogation through the monotone operator layer

Finally, a key challenge for any implicit model is to determine how to perform backpropagation through the layer. As with most implicit models, a potential benefit of the fixed-point conditions we describe is that, by using the implicit function theorem, it is possible to perform backpropagation without storing the intermediate iterates of the operator splitting algorithm in memory, and instead backpropagating directly through the equilibrium point.

To begin, we present a standard approach to differentiating through the fixed point $z^\star$ using the implicit function theorem. This formulation has some compelling properties for monDEQ, namely the fact that this (sub)gradient will always exist. When training a network via gradient descent, we need to compute the gradients of the loss function

$$\frac{\partial \ell}{\partial (\cdot)} = \frac{\partial \ell}{\partial z^\star} \frac{\partial z^\star}{\partial (\cdot)} \tag{8}$$

where $(\cdot)$ denotes some input to the layer or parameters, i.e. $W$, $x$, etc. The challenge here is computing (or left-multiplying by) the Jacobian $\partial z^\star / \partial (\cdot)$, since $z^\star$ is not an explicit function of the inputs. While it would be possible to simply compute gradients through the "unrolled" updates, e.g. $z^{k+1} = \sigma(Wz^k + Ux + b)$ for forward iteration, this would require storing each intermediate state $z^k$, a potentially memory-intensive operation. Instead, the following theorem gives an explicit formula for the necessary (sub)gradients. We state the theorem more directly in terms of the operators mentioned Theorem 1; that is, we use $\text{prox}_f^1(\cdot)$ in place of $\sigma(\cdot)$.

**Theorem 2.** *For the equilibrium point $z^\star = \text{prox}_f^1(Wz^\star + Ux + b)$, we have*

$$\frac{\partial \ell}{\partial (\cdot)} = \frac{\partial \ell}{\partial z^\star}(I - JW)^{-1}J\frac{\partial(Wz^\star + Ux + b)}{\partial (\cdot)} \tag{9}$$

*where*

$$J = \text{D} \, \text{prox}_f^1(Wz^\star + Ux + b) \tag{10}$$

*denotes the Clarke generalized Jacobian of the nonlinearity evaluated at the point $Wz^\star + Ux + b$. Furthermore, for the case that $(I - W) \succeq mI$, this derivative always exists.*

To apply the theorem in practice to perform reverse-mode differentiation, we need to solve the system

$$(I - JW)^{-T}\left(\frac{\partial \ell}{\partial z^\star}\right)^T. \tag{11}$$

The above system is a linear equation and while it is typically computationally infeasible to compute the inverse $(I - JW)^{-T}$ exactly, we could compute a solution to $(I - JW)^{-T}v$ using, e.g., conjugate gradient methods. However, we present an alternative formulation to computing (11) as the solution to a (linear) monotone operator splitting problem:

**Algorithm 3** Forward-backward equilibrium backpropagation

$u := 0;$  $\text{err} := 1;$  $v := \frac{\partial \ell}{\partial z^*}$
**while** err $> \epsilon$ **do**
    $u^+ := (1-\alpha)u + \alpha W^T u$
    $u_i^+ := \begin{cases} \frac{u_i^+ + \alpha v_i}{1 + \alpha(1 + D_{ii})} & \text{if } D_{ii} < \infty \\ 0 & \text{if } D_{ii} = \infty \end{cases}$
    $\text{err} := \frac{\|u^+ - u\|_2}{\|u^+\|_2}$
    $u := u^+$
**return** $u$

**Algorithm 4** Peaceman-Rachford equilibrium backpropagation

$z, u := 0;$  $\text{err} := 1;$  $v := \frac{\partial \ell}{\partial z^*};$  $V := (I + \alpha(I - W))^{-1}$
**while** err $> \epsilon$ **do**
    $u^{1/2} := 2z - u$
    $z^{1/2} := V^T u^{1/2}$
    $u^+ := 2z^{1/2} - u^{1/2}$
    $z_i^+ := \begin{cases} \frac{u_i^+ + \alpha v_i}{1 + \alpha(1 + D_{ii})} & \text{if } D_{ii} < \infty \\ 0 & \text{if } D_{ii} = \infty \end{cases}$
    $\text{err} := \frac{\|z^+ - z\|_2}{\|z^+\|_2}$
    $z, u := z^+, u^+$
**return** $z$

**Theorem 3.** *Let $z^\star$ be a solution to the monotone operator splitting problem defined in Theorem 1, and define $J$ as in (10). Then for $v \in \mathbb{R}^n$ the solution of the equation*

$$u^\star = (I - JW)^{-T} v \tag{12}$$

*is given by*

$$u^\star = v + W^T \tilde{u}^\star \tag{13}$$

*where $\tilde{u}^\star$ is a zero of the operator splitting problem $0 \in (\tilde{F} + \tilde{G})(u^\star)$, with operators defined as*

$$\tilde{F}(\tilde{u}) = (I - W^T)(\tilde{u}), \quad \tilde{G}(\tilde{u}) = D\tilde{u} - v \tag{14}$$

*where $D$ is a diagonal matrix defined by $J = (I + D)^{-1}$ (where we allow for the possibility of $D_{ii} = \infty$ for $J_{ii} = 0$).*

An advantage of this approach when using Peaceman-Rachford splitting is that it allows us to reuse a fast method for multiplying by $(I + \alpha(I - W))^{-1}$ which is required by Peaceman-Rachford during both the forward pass (equilibrium solving) and backward pass (backpropagation) of training a monDEQ. Algorithms detailing both the Peaceman-Rachford and forward-backward solvers for the backpropagation problem (14) are given in Algorithms 3 and 4.

## 4 Example monotone operator networks

With the basic foundations from the previous section, we now highlight several different instantiations of the monDEQ architecture. In each of these settings, as in Theorem 1, we will formulate the objective as one of finding a solution to the operator splitting problem $0 \in (F + G)(z^\star)$ for

$$F(z) = (I - W)(z) - (Ux + b), \quad G = \partial f \tag{15}$$

or equivalently as computing an equilibrium point $z^\star = \text{prox}_f^1(Wz^\star + Ux + b)$.

In each of these settings we need to define what the input and hidden state $x$ and $z$ correspond to, what the $W$ and $U$ operators consist of, and what is the function $f$ which determines the network nonlinearity. Key to the application of monotone operator methods are that 1) we need to constrain the $W$ matrix such that $I - W \succeq mI$ as described in the previous section and 2) we need a method to compute (or solve) the inverse $(I + \alpha(I - W))^{-1}$, needed e.g. for Peaceman-Rachford; while this would not be needed if using only forward-backward splitting, we believe that the full power of the monotone operator view is realized precisely when these more involved methods are possible.

### 4.1 Fully connected networks

The simplest setting, of course, is the case we have largely highlighted above, where $x \in \mathbb{R}^d$ and $z \in \mathbb{R}^n$ are unstructured vectors, and $W \in \mathbb{R}^{n \times n}$ and $U \in \mathbb{R}^{n \times d}$ and $b \in \mathbb{R}^n$ are dense matrices and vectors respectively. As indicated above, we parameterize $W$ directly by $A, B \in \mathbb{R}^{n \times n}$ as in (5). Since the $Ux$ term simply acts as a bias in the iteration, there is no constraint on the form of $U$.

We can form an inverse directly by simply forming and inverting the matrix $I + \alpha(I - W)$, which has cost $O(n^3)$. Note that this inverse needs to be formed only once, and can be reused over all iterations of the operator splitting method and over an entire batch of examples (but recomputed, of course, when $W$ changes). Any proximal function can be used as the activation: for example the ReLU, though as mentioned there are also close approximations to the sigmoid, tanh, and softplus.

## 4.2 Convolutional networks

The real power of monDEQs comes with the ability to use more structured linear operators such as convolutions. We let $x \in \mathbb{R}^{ds^2}$ be a $d$-channel input of size $s \times s$ and $z \in \mathbb{R}^{ns^2}$ be a $n$-channel hidden layer. We also let $W \in \mathbb{R}^{ns^2 \times ns^2}$ denote the linear form of a 2D convolutional operator and similarly for $U \in \mathbb{R}^{ns^2 \times ds^2}$. As above, $W$ is parameterized by two additional convolutional operators $A, B$ of the same form as $W$. Note that this implicitly increases the receptive field size of $W$: if $A$ and $B$ are $3 \times 3$ convolutions, then $W = (1 - m)I - A^T A + B - B^T$ will have an effective kernel size of 5.

**Inversion**   The benefit of convolutional operators in this setting is the ability to perform efficient inversion via the fast Fourier transform. Specifically, in the case that $A$ and $B$ represent circular convolutions, we can reduce the matrices to block-diagonal form via the discrete Fourier transform (DFT) matrix

$$A = F_s D_A F_s^*  \tag{16}$$

where $F_s$ denotes (a permuted form of) the 2D DFT operator and $D_A \in \mathbb{C}^{ns^2 \times ns^2}$ is a (complex) block diagonal matrix where each block $D_{Aii} \in \mathbb{C}^{n \times n}$ corresponds to the DFT at one particular location in the image. In this form, we can efficiently multiply by the inverse of the convolutional operator, noting that

$$
\begin{aligned}
I + \alpha(I - W) &= (1 + \alpha m)I + \alpha A^T A - \alpha B + \alpha B^T \\
&= F_s((1 + \alpha m)I + \alpha D_A^* D_A - D_B + D_B^*)F_s^*.
\end{aligned}
\tag{17}
$$

The inner term here is itself a block diagonal matrix with complex $n \times n$ blocks (each block is also guaranteed to be invertible by the same logic as for the full matrix). Thus, we can multiply a set of hidden units $z$ by the inverse of this matrix by simply inverting each $n \times n$ block, taking the fast Fourier transform (FFT) of $z$, multiplying each corresponding block of $F_s z$ by the corresponding inverse, then taking the inverse FFT. The details are given in Appendix C.

The computational cost of multiplying by this inverse is $O(n^2 s^2 \log s + n^3 s^2)$ to compute the FFT of each convolutional filter and precompute the inverses, and then $O(bns^2 \log s + bn^2 s^2)$ to multiply by the inverses for a set of hidden units with a minibatch of size $b$. Note that just computing the convolutions in a normal manner has cost $O(bn^2 s^2)$, so that these computations are on the same order as performing typical forward passes through a network, though empirically 2-3 times slower owing to the relative complexity of performing the necessary FFTs.

One drawback of using the FFT in this manner is that it requires that all convolutions be circular; however, this circular dependence can be avoided using zero-padding, as detailed in Section C.2.

## 4.3 Forward multi-tier networks

Although a single fully-connected or convolutional operator within a monDEQ can be suitable for small-scale problems, in typical deep learning settings it is common to model hidden units at different hierarchical levels. While monDEQs may seem inherently "single-layer," we can model this same hierarchical structure by incorporating structure into the $W$ matrix. For example, assuming a convolutional setting, with input $x \in \mathbb{R}^{ds^2}$ as in the previous section, we could partition $z$ into $L$ different hierarchical resolutions and let $W$ have a multi-tier structure, e.g.

$$
z = \begin{bmatrix} z_1 \in \mathbb{R}^{n_1 s_1^2} \\ z_2 \in \mathbb{R}^{n_2 s_2^2} \\ \vdots \\ z_L \in \mathbb{R}^{n_L s_L^2} \end{bmatrix}, \qquad
W = \begin{bmatrix} W_{11} & 0 & 0 & \cdots & 0 \\ W_{21} & W_{22} & 0 & \cdots & 0 \\ 0 & W_{32} & W_{33} & \cdots & 0 \\ \vdots & \vdots & \vdots & \ddots & \vdots \\ 0 & 0 & 0 & \cdots & W_{LL} \end{bmatrix}
$$

where $z_i$ denotes the hidden units at level $i$, an $s_i \times s_i$ resolution hidden unit with $n_i$ channels, and where $W_{ii}$ denotes an $n_i$ channel to $n_i$ channel convolution, and $W_{i+1,i}$ denotes an $n_i$ to $n_{i+1}$ channel, *strided* convolution. This structure of $W$ allows for both inter- and intra-tier influence.

One challenge is to ensure that we can represent $W$ with the form $(1 - m)I - A^T A + B - B^T$ while still maintaining the above structure, which we achieve by parameterizing each $W_{ij}$ block appropriately. Another consideration is the inversion of the multi-tier operator, which can be achieved via the FFT similarly as for single-convolutional $W$, but with additional complexity arising from the fact that the $A_{i+1,i}$ convolutions are strided. These details are described in Appendix D.

**CIFAR-10**

| Method | Model size | Acc. |
|---|---|---|
| Neural ODE | 172K | 55.3±0.3% |
| Aug. Neural ODE | 172K | 58.9±2.8% |
| Neural ODE[†*] | 1M | 59.9% |
| Aug. Neural ODE[†*] | 1M | 73.4% |
| monDEQ (ours) | | |
| Single conv | 172K | **74.0±0.1%** |
| Multi-tier | 170K | 72.0±0.3% |
| Single conv* | 854K | 82.0±0.3% |
| Multi-tier* | 1M | **89.0±0.3%** |

**SVHN**

| Method | Model size | Acc. |
|---|---|---|
| Neural ODE[‡] | 172K | 81.0% |
| Aug. Neural ODE[‡] | 172K | 83.5% |
| monDEQ (ours) | | |
| Single conv | 172K | 88.7±1.1% |
| Multi-tier | 170K | **92.4±0.1%** |

**MNIST**

| Method | Model size | Acc. |
|---|---|---|
| Neural ODE[‡] | 84K | 96.4% |
| Aug. Neural ODE[‡] | 84K | 98.2% |
| monDEQ (ours) | | |
| Fully connected | 84K | 98.1±0.1% |
| Single conv | 84K | **99.1±0.1%** |
| Multi-tier | 81K | 99.0±0.1% |

Table 1: Test accuracy of monDEQ models compared to Neural ODEs and Augmented Neural ODEs. *with data augmentation; [†]best test accuracy before training diverges; [‡]as reported in [10].

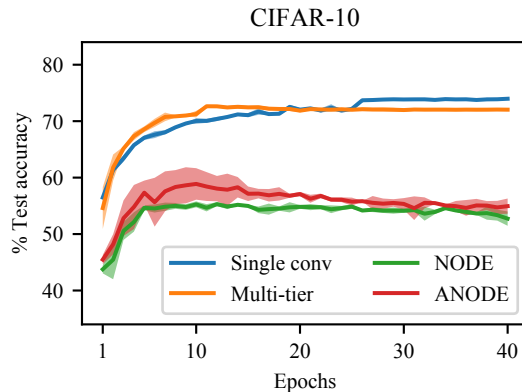

Figure 1: Test accuracy of monDEQs during training on CIFAR-10, with NODE [8] and ANODE [10] for comparison. NODE and ANODE curves obtained using code provided by [10].

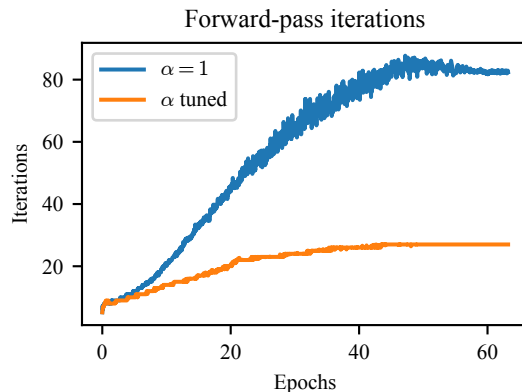

Figure 2: Iterations required by Peaceman-Rachford equilibrium solving over the course of training, for best $\alpha$ and $\alpha = 1$.

## 5 Experiments

To test the expressive power and training stability of monDEQs, we evaluate the monDEQ instantiations described in Section 4 on several image classification benchmarks. We take as a point of comparison the Neural ODE (NODE) [8] and Augmented Neural ODE (ANODE) [10] models, the only other implicit-depth models which guarantee the existence and uniqueness of a solution. We also assess the stability of training standard DEQs of the same form as our monDEQs.

The training process relies upon the operator splitting algorithms derived in Sections 3.4 and 3.5; for each batch of examples, the forward pass of the network involves finding the network fixed point (Algorithm 1 or 2), and the backward pass involves backpropagating the loss gradient through the fixed point (Algorithm 3 or 4). We analyze the convergence properties of both the forward-backward and Peaceman-Rachford operator splitting methods, and use the more efficient Peaceman-Rachford splitting for our model training. For further training details and model architectures see Appendix E. Experiment code can be found at http://github.com/locuslab/monotone_op_net.

**Performance on image benchmarks** We train small monDEQs on CIFAR-10 [17], SVHN [22], and MNIST [18], with a similar number of parameters as the ODE-based models reported in [8] and [10]. The results (averages over three runs) are shown in Table 1. Training curves for monDEQs, NODE, and ANODE on CIFAR-10 are show in Figure (1) and additional training curves are shown in Figure F1. Notably, except for the fully-connected model on MNIST, all monDEQs significantly outperform the ODE-based models across datasets. We highlight the performance of the small single convolution monDEQ on CIFAR-10 which outperforms Augmented Neural ODE by 15.1%.

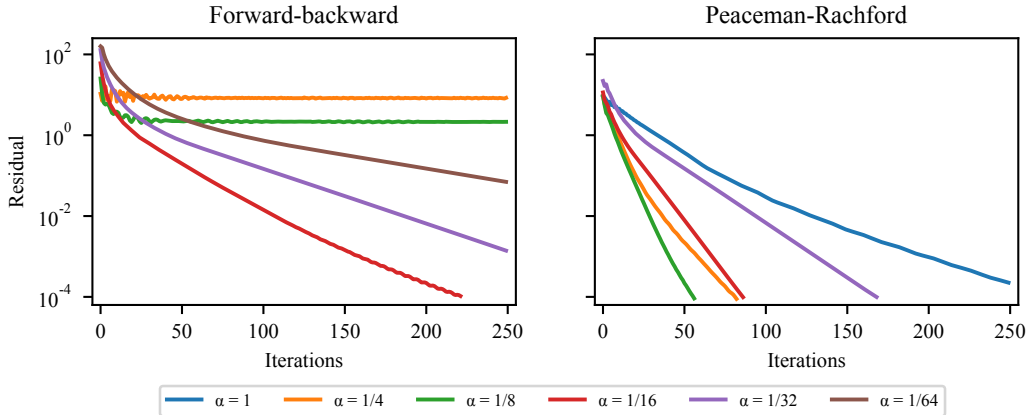

Figure 3: Convergence of Peaceman-Rachford and forward-backward equilibrium solving, on fully-trained model.

We also attempt to train standard DEQs of the same structure as our small multi-tier convolutional monDEQ. We train DEQs both with unconstrained $W$ and with $W$ having the monotone parameterization (5), and solve for the fixed point using Broyden's method as in [5]. All models quickly diverge during the first few epochs of training, even when allowed 300 iterations of Broyden's method.

Additionally, we train two larger monDEQs on CIFAR-10 with data augmentation. The strong performance (89% test accuracy) of the multi-tier network, in particular, goes a long way towards closing the performance gap with traditional deep networks. For comparison, we train larger NODE and ANODE models with a comparable number of parameters (~1M). These attain higher test accuracy than the smaller models during training, but diverge after 10-30 epochs (see Figure F1).

**Efficiency of operator splitting methods** We compare the convergence rates of Peaceman-Rachford and forward-backward splitting on a fully trained model, using a large multi-tier monDEQ trained on CIFAR-10. Figure 3 shows convergence for both methods during the forward pass, for a range of $\alpha$. As the theory suggests, the convergence rates depend strongly on the choice of $\alpha$. Forward-backward does not converge for $\alpha > 0.125$, but convergence speed varies inversely with $\alpha$ for $\alpha < 0.125$. In contrast, Peaceman-Rachford is guaranteed to converge for any $\alpha > 0$ but the dependence is non-monotonic. We see that, for the optimal choice of $\alpha$, Peaceman-Rachford can converge much more quickly than forward-backward. The convergence rate also depends on the Lipschitz parameter $L$ of $I - W$, which we observe increases during training. Peaceman-Rachford therefore requires an increasing number of iterations during both the forward pass (Figure 2) and backward pass (Figure F2).

Finally, we compare the efficiency of monDEQ to that of the ODE-based models. We report the time and number of function evaluations (OED solver steps or operator splitting iterations) required by the ~170k-parameter models to train on CIFAR-10 for 40 epochs. The monDEQ, neural ODE, and ANODE training takes respectively 1.4, 4.4, and 3.3 hours, with an average of 20, 96, and 90 function evals per minibatch. Note however that training the larger 1M-parameter monDEQ on CIFAR-10 requires 65 epochs and takes 16 hours. All experiments are run on a single RTX 2080 Ti GPU.

# 6 Conclusion

The connection between monotone operator splitting and implicit network equilibria brings a new suite of tools to the study of implicit-depth networks. The strong performance, efficiency, and guaranteed stability of monDEQ indicate that such networks could become practical alternatives to deep networks, while the flexibility of the framework means that performance can likely be further improved by, e.g. imposing additional structure on $W$ or employing other operator splitting methods. At the same time, we see potential for the study of monDEQs to inform traditional deep learning itself. The guarantees we can derive about what architectures and algorithms work for implicit-depth networks may give us insights into what will work for explicit deep networks.

## Broader impact statement

While the main thrust of our work is foundational in nature, we do demonstrate the potential for implicit models to become practical alternatives to traditional deep networks. Owing to their improved memory efficiency, these networks have the potential to further applications of AI methods on edge devices, where they are currently largely impractical. However, the work is still largely algorithmic in nature, and thus it is much less clear the immediate societal-level benefits (or harms) that could result from the specific tehniques we propose and demonstrate in this paper.

## Acknowledgements

Ezra Winston is supported by a grant from the Bosch Center for Artificial Intelligence.

## Footnotes

[1]We largely use the terms "fixed point" and "equilibrium point" interchangably in this work, using fixed point in the context of an iterative procedure, and equilibrium point to refer more broadly to the point itself.

[2]This setting can also be seen as corresponding to a recurrent network with identical inputs at each time (indeed, this is the view of so-called attractor networks [23]). However, because in modern usage recurrent networks typically refer to sequential models with *different* inputs at each time, we don't adopt this terminology.

[3]For non-symmetric matrices, which of course is typically the case with $W$, positive definiteness is defined as the positive definiteness of the symmetric component $I - W \succeq mI \Leftrightarrow I - (W + W^T)/2 \succeq mI$.

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
