[Supplementary Material]

# A  Monotone operator theory

We briefly review some of the basic properties of monotone operators that we make use of throughout this work. A *relation* or *operator* (which in our setting will often roughly correspond to a set-valued function), is a subset of the space $F \subseteq \mathbb{R}^n \times \mathbb{R}^n$; we use the notation $F(x) = \{y|(x,y) \in F\}$ or simply $F(x) = y$ if only a single $y$ is contained in this set. We make use of a few basic operators and relations: the identity operator $I = \{(x,x)|x \in \mathbb{R}^n\}$; the operator sum $(F + G)(x) = \{(x,y+z)|(x,y) \in F, (x,z) \in G\}$; the inverse operator $F^{-1}(x,y) = \{(y,x)|(x,y) \in F\}$; and the subdifferential operator $\partial f = \{(x, \partial f(x))| \in \mathrm{dom} f\}$. An operator $F$ has Lipschitz constant $L$ if for any $(x,u), (y,v) \in F$

$$\|u - v\|_2 \leq L\|x - y\|_2. \tag{A1}$$

An operator $F$ is monotone if

$$(u - v)^T(x - y) \geq 0, \quad \forall (x,u), (y,v) \in F \tag{A2}$$

which for the case of $F$ being a function $F : \mathbb{R}^n \to \mathbb{R}^n$ is equivalent to the condition

$$(F(x) - F(y))^T(x - y) \geq 0, \quad \forall x, y \in \mathrm{dom}\, F. \tag{A3}$$

In the case of scalar-valued functions, this corresponds to our common notion of a monotonic function. The operator $F$ is strongly monotone with parameter $m$ if

$$(u - v)^T(x - y) \geq m\|x - y\|^2, \quad \forall (x,u), (y,v) \in F. \tag{A4}$$

A monotone operator $F$ is *maximal monotone* if no other monotone operator strictly contains it; formally, most of the convergence properties we use require maximal monotonicity, though we are intentionally informal about this and merely use the established fact that several well-known operators are maximal monotone. Specifically, a linear operator $F(x) = Gx + h$ for $G \in \mathbb{R}^{n \times n}$ and $h \in \mathbb{R}^n$ is (maximal) monotone if and only if $G + G^T \succeq 0$ and strongly monotone if $G + G^T \succeq mI$. Similarly, a subdifferentiable operator $\partial f$ is maximal monotone iff $f$ is a convex closed proper (CCP) function.

The resolvent and Cayley operators for an operator $F$ are denoted $R_F$ and $C_F$ and respectively defined as

$$R_F = (I + \alpha F)^{-1}, \quad C_F = 2R_F - I \tag{A5}$$

for any $\alpha > 0$. The resolvent and Cayley operators are non-expansive (i.e., have Lipschitz constant $L \leq 1$) for any maximal monotone $F$, and are contractive (i.e. $L < 1$) for strongly monotone $F$.

We will mainly use two well-known properties of these operators. First, when $F(x) = Gx + h$ is linear, then

$$R_F(x) = (I + \alpha G)^{-1}(x - \alpha h) \tag{A6}$$

and when $F = \partial f$ for some CCP function $f$, then the resolvent is given by a proximal operator

$$R_F(x) = \mathrm{prox}_f^\alpha(x) \equiv \underset{z}{\mathrm{argmin}} \frac{1}{2}\|x - z\|_2^2 + \alpha f(z). \tag{A7}$$

Operator splitting approaches refer to methods to find a zero in a sum of operators (assumed here to be maximal monotone), i.e., find $x$ such that

$$0 \in (F + G)(x). \tag{A8}$$

There are many such operator splitting methods, which lead to different approaches in their application to our subsequent implicit networks, but the two we use mainly in this work are 1) *forward-backward* splitting, given by the update

$$x^{k+1} := R_G(x^k - \alpha F(x^k)); \tag{A9}$$

and 2) *Peaceman-Rachford* splitting, which is given by the iteration

$$u^{k+1} = C_F C_G(u^k), \quad x^k = R_G(u^k). \tag{A10}$$

Both methods will converge linearly to an $x$ that is a zero of the operator sum under certain conditions: a sufficient condition for forward-backward to converge is that $F$ be strongly monotone with parameter $m$ and Lipschitz with constant $L$ and $\alpha < 2m/L^2$; for Peaceman-Rachford, the method will converge for any choice of $\alpha$ for strongly monotone $F$, though the convergence speed will often vary substantially based upon $\alpha$.

# B  Proofs

## B.1  Proof of Theorem 1

*Proof.* The proof here is immediate: the forward-backward algorithm applied to the above operators with $\alpha = 1$ corresponds exactly to the network's fixed-point iteration:

$$
\begin{aligned}
z^{k+1} &= R_G(z^k - \alpha F(z^k)) \\
&= \mathrm{prox}_f^\alpha(z^k - \alpha(I - W)z^k + \alpha(Ux + b)) \\
&= \mathrm{prox}_f^1(Wz^k + Ux + b).
\end{aligned}
$$
$\square$

## B.2  Proof of Proposition 1

*Proof.* First assume $W$ is of this form. Then clearly

$$(I - W)/2 + (I - W)^T/2 = mI + A^T A \succeq mI. \tag{B1}$$

Alternatively, if $I - W \succeq mI \iff (1 - m)I \succeq (W + W^T)/2$, then

$$(W + W^T)/2 = (1 - m)I - A^T A. \tag{B2}$$

Thus

$$
\begin{aligned}
W &= (W + W^T)/2 + (W - W^T)/2 \\
&= (1 - m)I - A^T A + B - B^T.
\end{aligned}
$$
$\square$

## B.3  Proof of Theorem 2

*Proof.* Differentiating both sides of the fixed-point equation $z^\star = \sigma(Wz^\star + Ux + b)$ we have

$$
\begin{aligned}
\frac{\partial z^\star}{\partial(\cdot)} &= \frac{\partial \, \mathrm{prox}_f^1(Wz^\star + Ux + b)}{\partial(\cdot)} \\
&= J\left(W\frac{\partial z^\star}{\partial(\cdot)} + \frac{\partial(Wz^\star + Ux + b)}{\partial(\cdot)}\right)
\end{aligned} \tag{B3}
$$

for $J$ defined in (10) (we require the Clarke generalized Jacobian owing to the fact that the nonlinearity need not be a smooth function). Rearranging we get

$$
\begin{aligned}
(I - JW)\frac{\partial z^\star}{\partial(\cdot)} &= J\frac{\partial(Wz^\star + Ux + b)}{\partial(\cdot)} \\
\Leftrightarrow \frac{\partial z^\star}{\partial(\cdot)} &= (I - JW)^{-1}J\frac{\partial(Wz^\star + Ux + b)}{\partial(\cdot)}.
\end{aligned} \tag{B4}
$$

To show that this derivative always exists, we need to show that the $I - JW$ matrix is nonsingular. Owing to the fact that proximal operators are monotone and non-expansive, we have $0 \leq J_{ii} \leq 1$. First, letting $\lambda(\cdot)$ denote the set of eigenvalues of a matrix, note that

$$\lambda(I - JW) = \lambda(I - J^{1/2}WJ^{1/2}). \tag{B5}$$

This follows from the similarity transform $\lambda(I - JW) = \lambda(J^{-1/2}(I - JW)J^{1/2})$ for $J > 0$ and the case of $J_{ii} = 0$ follows via the continuity of eigenvalues taking $\lim J_{ii} \to 0$. Now, using the fact that $0 \preceq J \preceq I$, we have

$$
\begin{aligned}
&\mathrm{Re}\ \lambda(I - J^{1/2}WJ^{1/2}) \\
&= \mathrm{Re}\ \lambda(I - J + J^{1/2}(I - W)J^{1/2}) > 0
\end{aligned} \tag{B6}
$$

since $I - W \succeq mI$ and $I - J \succeq 0$. $\square$

## B.4  Proof of Theorem 3

*Proof.* We begin with the case where $J_{ii} \neq 0$ and thus $D_{ii} < \infty$. As above, because proximal operators are themselves monotone non-expansive operators, we always $0 \leq J_{ii} \leq 1$, so that $D_{ii} \geq 0$.

Now, first assuming that $J_{ii} > 0$, and hence $D_{ii} < \infty$, we have

$$
\begin{aligned}
&u = (I - JW)^{-T}v \\
&\Leftrightarrow (I - W^T(I + D)^{-1})u = v \\
&\Leftrightarrow W^{-T}u - (I + D)^{-1}u = W^{-T}v \\
&\Leftrightarrow (I + D)W^{-T}u - u = (I + D)W^{-T}v \\
&\Leftrightarrow W^{-T}u - u + DW^{-T}u = (I + D)W^{-T}v \\
&\Leftrightarrow \tilde{u} - W^T\tilde{u} + D\tilde{u} = (I + D)W^{-T}v
\end{aligned}
\tag{B7}
$$

where we define $\tilde{u} = W^{-T}u$. To simplify the right hand side of this equation and remove the explicit $W^{-T}v$ terms[4] we note that

$$
(I - JW)^{-T} = (I - W^TJ)^{-1} = I + (I - W^TJ)^{-1}W^TJ. \tag{B8}
$$

Thus, we can always solve the above equation with the $v$ term of the form $W^TJv$, giving

$$
(I + D)W^{-T}W^TJv = (I + D)Jv = v. \tag{B9}
$$

This gives us a (linear) operator splitting problem with the $\tilde{F}$ and $\tilde{G}$ operators given in (14).

To handle the case where $J_{ii} = 0 \Leftrightarrow D_{ii} = \infty$, we can simply take the limit $D_{ii} \to \infty$, and note that all the operators are well-defined for this case. For instance, the resolvent operator

$$
R_{\tilde{G}}(u) = (I + \alpha(I + D))^{-1}(u + \alpha v) \tag{B10}
$$

and thus

$$
R_{\tilde{G}}(u)_{ii} = \frac{u + \alpha v}{1 + \alpha(1 + D_{ii})} \to 0 \tag{B11}
$$

as $D_{ii} \to \infty$.

Finally, owing to the fact that $I - W^T \succeq mI$ and $D_{ii} \geq 0$, the $\tilde{F}$ and $\tilde{G}$ operators are strongly monotone and monotone respectively, we conclude that operator splitting techniques applied to the problem will be guaranteed to converge. $\qquad\square$

## C  Convolutional monDEQs

### C.1  Inversion via the discrete Fourier transform

First consider the case where $W \in \mathbb{R}^{s^2 \times s^2}$ is the matrix representation of an unstrided (circular) convolution with a single input channel and single output channel. The convolution operates on vectorized $s \times s$ inputs. It is well known that $W$ is diagonalized by the 2D DFT operator $\mathscr{F}_s = F_s \otimes F_s$ where $F_s$ is the Fourier basis matrix $(F_s)_{ij} = \frac{1}{s}\omega^{(i-1)(j-1)}$ and $\omega = \exp(2\pi\iota/s)$. We denote $\iota = \sqrt{-1}$ to avoid confusion with the index $i$. So

$$
\mathscr{F}_s W \mathscr{F}_s^* = D, \tag{C1}
$$

a complex diagonal matrix.

Now take the case where $W \in \mathbb{R}^{ns^2 \times ns^2}$ has $n$ input and output channels. Then

$$
(I_n \otimes \mathscr{F}_s)W(I_n \otimes \mathscr{F}_s^*) = D = \begin{bmatrix} D_{11} & D_{12} & \cdots & D_{1n} \\ D_{21} & D_{22} & \cdots & D_{2n} \\ \vdots & \vdots & \ddots & \vdots \\ D_{n1} & D_{n2} & \cdots & D_{nn} \end{bmatrix} \tag{C2}
$$

where $I_n$ is the $n \times n$ identity matrix and each block $D_{ij} \in \mathbb{C}^{s^2 \times s^2}$ is a complex diagonal matrix. We will denote $\mathscr{F}_{s,n} = I_n \otimes \mathscr{F}_s$.

It is more efficient to consider the permuted form of $D$

$$\hat{D} = \begin{bmatrix} \hat{D}^1 & 0 & \cdots & 0 \\ 0 & \hat{D}^2 & \cdots & 0 \\ \vdots & \vdots & \ddots & \vdots \\ 0 & 0 & \cdots & \hat{D}^{s^2} \end{bmatrix} \tag{C3}$$

where each block $\hat{D}^k \in \mathbb{C}^{n \times n}$, consists of the $k$th diagonal elements of all the $D_{ij}$, that is $\hat{D}^k_{ij} = (D_{ij})_{kk}$. Then inverting or multiplying by $\hat{D}$ reduces to inverting or multiplying by the diagonal blocks, which is amenable to accelerated batch-wise computation in the form of an $s^2 \times n \times n$ tensor. However, the original form (C2) is more convenient mathematically and we use that here.

To perform the required inversion of the operator

$$I + \alpha(I - W) = (1 + \alpha m)I + \alpha A^T A - \alpha B + \alpha B^T \tag{C4}$$

we use the fact that $\mathscr{F}_{s,n}$ is unitary and obtain

$$(1 + \alpha m)I + \alpha A^T A - \alpha B + \alpha B^T$$
$$= (1 + \alpha m)\mathscr{F}^*_{s,n}\mathscr{F}_{s,n} + \mathscr{F}^*_{s,n}(\alpha D^*_A \mathscr{F}_{s,n}\mathscr{F}^*_{s,n}D_A - D_B + D^*_B)\mathscr{F}_{s,n} \tag{C5}$$
$$= \mathscr{F}^*_{s,n}((1 + \alpha m)I + \alpha D^*_A D_A - D_B + D^*_B)\mathscr{F}_{s,n}.$$

The inner term here itself has the blockwise-diagonal form (C2). Thus, we can multiply a set of hidden units $z$ by the inverse of this matrix by considering the permuted form (C3), inverting each block $\hat{D}^i$, taking the FFT of $z$, multiplying each corresponding block of $\mathscr{F}_{s,n}z$ by the corresponding inverse, then taking the inverse FFT.

## C.2 Zero padding

One drawback to the above method is that using the FFT in this manner requires that all convolutions be circular. While empirically there is little drawback to simply replacing traditional convolutions with their circular variants, in some cases it may be desirable to avoid this setting, where information about the image may wrap around the borders. If it is desirable to avoid this, we explicitly remove any circular dependence by zero-padding the hidden units with $(k-1)/2$ border pixels, where $k$ denotes the receptive field size of the convolution. This zero padding can then be enforced by simply setting all the border entries to zero within the *nonlinearity* of the network; because setting an element to zero is equivalent to the proximal operator for the indicator of the zero set, such operations still fit within the monotone operator setting.

# D Multi-tier monDEQs

## D.1 Parameterization

Recall the setting of Section 4.3, with

$$z = \begin{bmatrix} z_1 \in \mathbb{R}^{n_1 s_1^2} \\ z_2 \in \mathbb{R}^{n_2 s_2^2} \\ \vdots \\ z_L \in \mathbb{R}^{n_L s_L^2} \end{bmatrix}, \qquad W = \begin{bmatrix} W_{11} & 0 & 0 & \cdots & 0 \\ W_{21} & W_{22} & 0 & \cdots & 0 \\ 0 & W_{32} & W_{33} & \cdots & 0 \\ \vdots & \vdots & \vdots & \ddots & \vdots \\ 0 & 0 & 0 & \cdots & W_{LL} \end{bmatrix}. \tag{D1}$$

To ensure $W$ has the form $(1 - m)I - A^T A + B - B^T$, we restrict both $A$ and $B$ to have the same bidiagonal structure as $W$. Then the diagonal terms $W_{ii}$ have the form

$$W_{ii} = (1 - m)I - A^T_{ii}A_{ii} - A^T_{i+1,i}A_{i+1,i} + B_{ii} - B^T_{ii} \tag{D2}$$

for $i < L$ and

$$W_{LL} = (1 - m)I - A^T_{LL}A_{LL} + B_{LL} - B^T_{LL}. \tag{D3}$$

To compute the off-diagonal terms $W_{i+1,i}$ note that restricting $W$ to be bidiagonal makes the off-diagonal terms of $B$ redundant. E.g. since $W_{12} = 0$, then

$$-A_{21}^T A_{22} - B_{21}^T = W_{12} = 0$$
$$\Rightarrow W_{21} = -A_{22}^T A_{21} + B_{21} = -2A_{22}^T A_{21}. \tag{D4}$$

## D.2   Inversion via the discrete Fourier transform

Consider $W$ of the form (D1) with convolutions

$$W_{ii} = (1-m)I - A_{ii}^T A_{ii} - A_{i+1,i}^T A_{i+1,i} + B_{ii} - B_{ii}^T$$
$$W_{LL} = (1-m)I - A_{LL}^T A_{LL} + B_{LL} - B_{LL}^T \tag{D5}$$
$$W_{i+1,i} = -2A_{i+1,i+1}^T A_{i+1,i}.$$

Here the $A_{ii}$ and $B_{ii}$ terms are unstrided convolutions with $n_i$ input and $n_i$ output channels, while the $A_{i,i+1}$ are strided convolutions with $n_i$ input channels and $n_{i+1}$ output channels.

In order to multiply by $(I + \alpha(I - W))^{-1}$, we use back substitution to solve for $x$ in

$$z = (I + \alpha(I - W))x. \tag{D6}$$

Let $W' = (I + \alpha(I - W))$. The back substitution proceeds by tiers, i.e.

$$x_1 = W_{11}'^{-1} z_1$$
$$x_2 = W_{22}'^{-1}(z_2 - W_{21}' x_1)$$
$$x_3 = W_{33}'^{-1}(z_3 - W_{32}' x_2) \tag{D7}$$
$$\vdots$$

Therefore only the diagonal blocks $W_{ii}'$ need be inverted. The inversion of e.g.

$$W_{11}' = (1 + \alpha m)I + \alpha(A_{11}^T A_{11} + A_{21}^T A_{21} + B_{11} - B_{11}^T) \tag{D8}$$

is complicated by the fact that $A_{21}$ is strided, so that it is no longer diagonalized by the DFT. Instead, we perform inversion using the following proposition.

**Proposition D1.** *Let $A \in \mathbb{R}^{n_1 s^2 \times n_1 s^2}$ be an unstrided circular convolution with $n_1$ input and $n_1$ output channels, and $B \in \mathbb{R}^{n_2 s^2 \times n_1 s^2}$ a strided circular convolution with $n_1$ input and $n_2$ output channels and stride $r$ where $r$ divides $s$. Then*

$$(A + B^T B)^{-1} = \mathscr{F}_{s,n_1}^*(D_A^{-1} - D_A^{-1} D_B^*(I_{n_2} \otimes K)D_B D_A^{-1})\mathscr{F}_{s,n_1} \tag{D9}$$

*where*

$$D_A = \mathscr{F}_{s,n_1} A \mathscr{F}_{s,n_1}^*, \quad D_B = \mathscr{F}_{s,n_2} B \mathscr{F}_{s,n_1}^*,$$
$$K = S^T J(s^2 r^2 I + J^T S D_B D_A^{-1} D_B^* S^T J)^{-1} J^T S \tag{D10}$$

*where $J = 1_{r^2} \otimes I_{s^2/r^2}$ is $r^2$ stacked identity matrices of size $(s^2/r^2) \times (s^2/r^2)$ and $S = (I_r \otimes S_{s/r,s})$ is a permutation matrix where $S_{a,b} \in \mathbb{R}^{ab \times ab}$ denotes the perfect shuffle matrix defined by subselecting rows of the identity matrix $I_{ab}$, here given in MATLAB notation:*

$$S_{a,b} = \begin{bmatrix} I_{ab}(1:b:ab,:) \\ I_{ab}(2:b:ab,:) \\ \vdots \\ I_{ab}(b:b:ab,:) \end{bmatrix}. \tag{D11}$$

*Proof.* We will show that

$$A + B^T B = \mathscr{F}_{s,n_1}^*(D_A + D_B^*(I_{n_2} \otimes (\frac{1}{s^2 r^2} S^T J J^T S))D_B)\mathscr{F}_{s,n_1}. \tag{D12}$$

The desired result then follows by applying the Woodbury matrix idenetity.

We start by breaking $B$ into an unstrided convolution $B'$ which can be diagonalized by the DFT and a matrix $U_{r,s}$ which performs the striding on each channel:

$$B = (I_{n_2} \otimes U_{r,s})B' = (I_{n_2} \otimes U_{r,s})\mathscr{F}_{s,n_2}^* D_B \mathscr{F}_{s,n_1} \tag{D13}$$

where $U_{r,s} \in \mathbb{R}^{(s^2/r^2) \times s^2}$ is defined by subselecting rows of the identity matrix:

$$U_{r,s} = \begin{bmatrix} I_{s^2}(1:r:s,:) \\ I_{s^2}(rs+1:r:(r+1)s,:) \\ I_{s^2}(2rs+1:r:(2r+1)s,:) \\ \vdots \\ I_{s^2}(s^2-sr+1:r:s^2-s(r-1),:) \end{bmatrix}. \tag{D14}$$

So

$$B^T B = \mathscr{F}_{s,n_1}^* D_B^* (I_{n_2} \otimes (\mathscr{F}_s U_{r,s}^T U_{r,s} \mathscr{F}_s^*)) D_B \mathscr{F}_{s,n_1}. \tag{D15}$$

We want to show that $\mathscr{F}_s U_{r,s}^T U_{r,s} \mathscr{F}_s^* = \frac{1}{s^2 r^2} S^T J J^T S$. Observe that

$$U_{r,s}^T U_{r,s} = (T_{r,s} \otimes T_{r,s}) \tag{D16}$$

where $T_{r,s} \in \mathbb{R}^{s \times s}$ is given by

$$(T_{r,s})_{ij} = \begin{cases} 1 & \text{if } i = j \text{ and } i \pmod{r} = 1, \\ 0 & \text{else.} \end{cases} \tag{D17}$$

Then by the properties of Kronecker product

$$\mathscr{F}_s U_{r,s}^T U_{r,s} \mathscr{F}_s^* = (F_s \otimes F_s)(T_{r,s} \otimes T_{r,s})(F_s^* \otimes F_s^*) = (F_s T_{r,s} F_s^*) \otimes (F_s T_{r,s} F_s^*). \tag{D18}$$

We now show that $(F_s T_{r,s} F_s^*) = L$ where

$$L_{ij} = \begin{cases} \frac{1}{sr} & \text{if } i \equiv j \pmod{s/r}, \\ 0 & \text{else.} \end{cases} \tag{D19}$$

To do so we use several properties of the roots of unity $z^k = \exp(2\pi\iota k/s)$.

1. If $a \equiv b \pmod{s}$ then $z^a = z^b$.
2. If $z$ is a primitive $s$th root of unity then $z^m$ is a primitive $a$th root of unity where $a = \frac{s}{\gcd(m,s)}$.
3. The sum of the $s$th roots of unity $\sum_{k=0}^{s-1} z^k = 0$ if $s > 1$.

We first compute $L_{ij}$ for the case when $i \equiv j \pmod{s/r}$, or in other words $i = j + \frac{ks}{r}$ for some integer $k$. We have

$$\begin{aligned} L_{ij} &= \frac{1}{s^2} \sum_{p=1:r:s} \omega^{(i-1)(p-1)} \bar{\omega}^{(p-1)(j-1)} \\ &= \frac{1}{s^2} \sum_{p=0:r:s-1} \exp(2\pi\iota p(i-j)/s) \\ &= \frac{1}{s^2} \sum_{p=0:r:s-1} \exp(2\pi\iota pk/r) \\ &= \frac{1}{s^2} \sum_{p=0}^{\frac{s}{r}-1} \exp(2\pi\iota pk) \\ &= \frac{1}{sr}. \end{aligned} \tag{D20}$$

For the case when $i \not\equiv j \pmod{s/r}$, or in other words $i = j + \frac{ks}{r} + m$ for some integers $k$ and $m$ with $-\frac{s}{r} < m < \frac{s}{r}$, we have

$$
\begin{aligned}
L_{ij} &= \frac{1}{s^2} \sum_{p=1:r:s} \omega^{(i-1)(p-1)} \bar{\omega}^{(p-1)(j-1)} \\
&= \frac{1}{s^2} \sum_{p=0:r:s-1} \exp(2\pi\iota p(i-j)/s) \\
&= \frac{1}{s^2} \sum_{p=0}^{\frac{s}{r}-1} \exp(2\pi\iota p(i-j)r/s) \\
&= \frac{1}{s^2} \sum_{p=0}^{\frac{s}{r}-1} \exp(2\pi\iota pmr/s)\exp(2\pi\iota pk).
\end{aligned}
\tag{D21}
$$

By property (2), since $\exp(2\pi\iota r/s)$ is a primitive $\frac{s}{r}$th root of unity, then $\exp(2\pi\iota mr/s)$ is a primitive $d$th root of unity where $d = \frac{s/r}{\gcd(m,s/r)}$. Since $d$ divides $s/r$, we can split the sum into several sums of $d$th roots of unity using property (1), each of which will sum to zero by property (3).

$$
\begin{aligned}
L_{ij} &= \frac{1}{s^2} \sum_{p=0}^{\frac{s}{r}-1} \exp(2\pi\iota pmr/s) \\
&= \frac{1}{s^2} \sum_{q=0}^{\frac{s}{rd}-1} \sum_{p=0}^{d-1} \exp(2\pi\iota(p+qd)mr/s) \\
&= \frac{1}{srd} \sum_{p=0}^{d-1} \exp(2\pi\iota pmr/s) \\
&= 0
\end{aligned}
\tag{D22}
$$

where the second equality follows from property (1) since $p = p + qd \pmod d$ and each sum in the third line is zero by property (3) since $\exp(2\pi\iota mr/s)$ is a primitive $d$th root of unity.

We now have $\mathscr{F}_s U_{r,s}^T U_{r,s} \mathscr{F}_s^* = L \otimes L$ and it remains to use properties of Kronecker product to show that $L \otimes L = \frac{1}{s^2 r^2} S^T J J^T S$. In particular we need associativity and the fact that for $A \in \mathbb{R}^{n \times n}$ and $B \in \mathbb{R}^{m \times m}$, we have

$$
B \otimes A = S_{n,m}(A \otimes B)S_{n,m}^T
\tag{D23}
$$

where $S_{n,m}$ is the perfect shuffle matrix. Note that $L = \frac{1}{sr} 1_{r \times r} \otimes I_{s/r}$ where $1_{r \times r}$ is the $r \times r$ matrix of all ones. Then

$$
\begin{aligned}
L \otimes L &= \frac{1}{s^2 r^2}(1_{r \times r} \otimes I_{s/r}) \otimes (1_{r \times r} \otimes I_{s/r}) \\
&= \frac{1}{s^2 r^2} 1_{r \times r} \otimes (I_{s/r} \otimes (1_{r \times r} \otimes I_{s/r})) \\
&= \frac{1}{s^2 r^2} 1_{r \times r} \otimes (S_{s/r,s}((1_{r \times r} \otimes I_{s/r}) \otimes I_{s/r})S_{s/r,s}^T) \\
&= \frac{1}{s^2 r^2} 1_{r \times r} \otimes (S_{s/r,s}(1_{r \times r} \otimes I_{s^2/r^2})S_{s/r,s}^T) \\
&= \frac{1}{s^2 r^2}(I_r \otimes S_{s/r,s})(1_{r^2 \times r^2} \otimes I_{s^2/r^2})(I_r \otimes S_{s/r,s}^T) \\
&= \frac{1}{s^2 r^2} S J J^T S^T
\end{aligned}
\tag{D24}
$$

which completes the proof. $\qquad\square$

|  | CIFAR-10 | | | |
|---|---|---|---|---|
|  | Single conv | Multi-tier | Single conv lg. | Multi-tier lg. |
| Num. channels | 81 | (16,32,60) | 200 | (64,128,128) |
| Num. params | 172,218 | 170,194 | 853,612 | 1,014,546 |
| Epochs | 40 | 40 | 65 | 65 |
| Initial lr | 0.001 | 0.01 | 0.001 | 0.001 |
| Lr schedule | step decay | step decay | 1-cycle | 1-cycle |
| Lr decay steps | 25 | 10 | - | - |
| Lr decay factor | 10 | 10 | - | - |
| Max learning rate | - | - | 0.01 | 0.05 |
| Data augmentation | - | - | ✓ | ✓ |

|  | SVHN | | MNIST | | |
|---|---|---|---|---|---|
|  | Single conv | Multi-tier | FC | Single conv | Multi-tier |
| Num. channels | 81 | (16,32,60) | 87* | 54 | (16, 32, 32) |
| Num. params | 172,218 | 170,194 | 84,313 | 84,460 | 81,394 |
| Initial lr | 0.001 | 0.001 | 0.001 | 0.001 | 0.001 |
| Epochs | 40 | 40 | 40 | 40 | 40 |
| Lr decay steps | 25 | 10 | 10 | 10 | 10 |
| Lr decay factor % | 10 | 10 | 10 | 10 | 10 |

Table E1: Model hyperparameters. *FC is a dense layer with output dimension of 87.

# E  Experiment details

## E.1  Model architecture

Recall that a monDEQ is defined by a choice of linear operators $W$ and $U$, bias $b$, and nonlinearity $\sigma$, and that we parameterize $W$ via linear operators $A$ and $B$. For all experiments we use $\sigma =$ ReLU. In the fully-connected network $A, B$ and $U$ are dense matrices; in the single-convolution network they are unstrided convolutions with kernel size 3. The structure of the multi-tier network is as described in (D1) and (D5); we use three tiers with unstrided convolutions for $U$ and $A_{ii}, B_{ii}$ and stride-2 convolutions for the subdiagonal terms $A_{i,i+1}$, all with kernels of size 3. The number of channels for single and multi-tier convolutional models varies by dataset, as shown in Table E1.

For all models, the fixed point $z^{\star}$ is mapped to logits $\hat{y}$ via a dense output layer, and the single convolution model first applies $4{\times}4$ average pooling:

$$\hat{y} = W_o z^{\star} + b_o \quad \text{or} \quad \hat{y} = W_o \, \text{AvgPool}_{4\times4}(z^{\star}) + b_o.$$

## E.2  Training details

Because $W = (1-m)I - A^T A + B - B^T$ contains both linear and quadratic terms, we find that a variant of weight normalization helps to keep the gradients of the different parameters on the same scale. For example, when $W$ is a dense matrix, we reparameterize $A^T A$ as $g\frac{A^T A}{\|A\|^2}$ and $B$ as $h\frac{B}{\|B\|}$, where $g$ and $h$ are learned scalars. When $W$ consists of a single or multi-tiered convolutions, we reparameterize each convolution kernel analogously.

All models are trained by running Peaceman-Rachford with error tolerence $\epsilon =$1e-2, which reduces the number of iterations without impacting performance. The monotonicity parameter $m$ also affects convergence speed since it controls the contraction factor of the relevant operators; consistent with this, we find that Peaceman-Rachford takes longer to converge for smaller $m$, and use $m = 1$ for all models since model performance is not sensitive to $m \in [0.01, 1]$. We also find that the Lipschitz parameter $L$ of $I - W$ increases during training, changing the optimal $\alpha$ value. We therefore tune $\alpha \in \{1, 1/2, 1/4, \ldots\}$ over the course of training so as to minimize forward-pass iterations.

One detail about stopping criteria for the splitting method: computing the residual $\|z^{k+1} - f(z^{k+1})\|/\|z^{k+1}\|$ requires an additional call to the function $f$. Therefore during training we instead use the criterion $\|z^{k+1} - z^k\|/\|z^{k+1}\| \le \epsilon$. The error shown in Figure 3 is the former, while the stopping criterion used in Figures 2 and F2 is the latter. Technically this latter criterion itself depends on both $\alpha$ and $L$; for different $\alpha$ and $L$ values, having $\|z^{k+1} - z^k\|/\|z^{k+1}\| \le \epsilon$ implies different bounds on the residual. However, we find that this effect is minimal, so that both stopping criteria work equally well in practice.

| | Train examples | Test examples | Image dim. | Num. channels |
|---|---|---|---|---|
| MNIST | 60,000 | 10,000 | $28 \times 28$ | 1 |
| SVHN | 73,257 | 26,032 | $32 \times 32$ | 3 |
| CIFAR-10 | 50,000 | 10,000 | $32 \times 32$ | 3 |

Table E2: Dataset statistics

Table E1 gives details of the training hyperparameters used for each model. All models are trained with ADAM [16], using batch size of 128. For all but the large CIFAR-10 models, the initial learning rate is chosen from {1e-2, 1e-3} and decayed by a factor of 10 after every 10 or 25 epochs, and the default ADAM momentum parameters are used. All training data is normalized to mean $\mu = 0$, standard deviation $\sigma = 1$.

**CIFAR-10 with data augmentation**   When training large models on CIFAR-10 we use standard data augmentation, consisting of zero-padding the $32 \times 32$ images to $40 \times 40$, then randomly cropping back to 32x32, and finally performing random horizontal flips. In order to reduce the number of training epochs, we use a single cycle of increasing and decreasing learning rate to achieve super-convergence [27]. The learning rate is increased from 1e-3 to the max learning rate (see Table E1) over 30 epochs, then decreased back to 1e-3 over 30 epochs, then held at 1e-3 for 5 epochs. (The max learning rate is chosen by training for a single epoch while increasing the learning rate until the loss diverges.) The momentum is also decreased from 0.95 to 0.85 over 30 epochs, then back to 0.95 over 30 epochs, then held at 0.95 for 5 epochs. However, we note that the model obtains the same performance when trained with constant learning rate of 1e-3 for around 200 epochs.

### E.3   Dataset statistics

MNIST [18] consists of black and white examples of handwritten digits 0-9. SVHN [22] consists of color images of digits 0-9 extracted from house numbers captured by Google Stree View. CIFAR-10 [17] consists of small images from 10 object classes. Dataset statistics are shown in Table E2.

# F   Additional results and figures

Figure F1: Test accuracy of monDEQs and Neural ODE models during training.

Figure F2: Iterations required by Peaceman-Rachford backprop over the course of training.

## Footnotes

[4]Although we could solve this operator splitting problem directly, the presence of the $W^{-T}v$ term has two notable downsides: 1) even if the $W$ matrix itself is nonsingular, it may be arbitrarily close to a singular matrix, thus making direct solutions with this matrix introduce substantial numerical errors; and 2) for operator splitting methods that do *not* require an inverse of $W$ (e.g. forward-backward splitting), it would be undesirable to require an explicit inverse in the backward pass.