[Reviews · NeurIPS 2020]

Review 1

Summary and Contributions: DEQ is a very interesting work, which models infinite depth with a constant memory. DEQ does not save the intermediate hidden layers and updates the weights with only the fixed point. However, it did not solve the problem that does the fixed point stable and unique? The paper used the theory of monotone operators and presented a solution to this problem.

Strengths: The paper used the theory of monotone operators and presented a practical solution to obtain unique fixed points.

Weaknesses: Compared with DEQ, the fixed points of MOE are much more stable and unique, which needs experiments to justify.

Correctness: Yes. Yes.

Clarity: Yes

Relation to Prior Work: Yes

Reproducibility: Yes

Additional Feedback: After reading the reviews and the rebuttal, I will keep my score (here is 7). DEQ is an important work and the presented extension is promising. The authors claimed in the rebuttal that they will add the comparisons with DEQ. However, it is not enough for me. For example, they need to investigate whether the more stable of the fixed point the better the test performance. Since it is an extension of DEQ, the paper should test the advantages over DEQ thoroughly. This prevents me from raising my score further. ------------------------------------------------------------------ Overall, the paper is well-written and is easy to follow. I have the following comments. 1. The paper argued that DEQ requires careful initialization and regularization and MOE does not require much tuning. However, the paper failed to verify this claim. Since MOE is the improvement of DEQ, the extensive comparison between the two methods is need. For example, compared with DEQ, the fixed points of MOE are much more stable and unique, which needs experiments to justify. 2. In Theorem 1, the function f should be convex closed proper (CCP). The paper needs to highlight the requirement. All non-decreasing activation functions can be represented as proximal operators. However, the function f in the proximal operators may be not CCP. The paper needs more discussions on this issue. 3. In Theorem 1, A and B are operators, while they are matrices in Proposition 1. It is better to use other letters in Proposition 1. 4. In line 399 of the Appendix, A should be G.


Review 2

Summary and Contributions: The paper develops a principled approach for training fixed-point networks. Given input $x$, fixed point networks define the output of a network as a fixed point $z$ of a computational block $f(x, z)$. The authors consider computational blocks consisting of a linear operator and a component-wise non-linearity; for such blocks, the authors propose a parametrization for the block to ensure a sufficient condition for the existence of a fixed point. Then the authors adopt two different operator splitting algorithms for the forward and backward passes of the fixed-point network. For the evaluation, the authors develop several architectures and compare them against Neural ODE. With the same parameter count, the proposed approach outperforms Neural ODE in terms of accuracy and computational efficiency. Additionally, the authors show the role of the fixed point solver hyperparameters and the scalability to more parameters.

Strengths: The original work on deep equilibrium models (DEQ) did not give any convergence guarantees for the underlying iterative process. The paper fills the gap with an alternative parametrization, an elegant and easy to implement solution. Similarly to DEQ, Monotone networks use implicit differentiation and do not store the intermediate steps of the underlying solver. As a result, the model training loop is more memory-efficient compared to conventional deep architectures. Another appealing feature is that the considered class of fixed-point networks include a variety of non-linear activations and, as the authors show, is extendable beyond fully-connected matrices. Monotone networks outperform NeuralODE (another implicit depth architecture that is, unlike DEQ, well-defined) with a similar number of parameters and demonstrate a room for further performance improvement.

Weaknesses: Compared to DEQ, Monotone Networks consider a less versatile class of functions (a price to pay for the convergence?). In particular, NeuralODE and DEQ extend to series data, but it is not clear how to extend MON to such data.

Correctness: The method and claims are correct. In the experimental section, it would be nice to see the results of NODE and ANODE for bigger model size along with the results of MON.

Clarity: The paper is very well written.

Relation to Prior Work: The authors clearly discuss the relation to DEQ and NeuralODE. Could the authors comment on how their approach compares to the "Implicit Deep Learning" paper?

Reproducibility: Yes

Additional Feedback: [Post rebuttal edit: Thank you for the thorough feedback. After the rebuttal, I am keeping my original score.] Is there any intuition on whether such architectures are suitable for regression tasks?


Review 3

Summary and Contributions: Authors have used the theory of monotone operators to develop a novel implicit-depth model. Unlike previous approaches such as NODE and ANODE, authors show that their proposed approach has stable convergence.

Strengths: The major impressive strength of the paper is the fact that the proposed approach significantly outperforms state-of-the-art results from NODE and ANODE papers.

Weaknesses: It is great that authors have compared against NODE and ANODE. However, there are still some comparisons that are missing compared to the reported results in NODE and ANODE. One more comment about simulation results is that it would be great if authors could report the standard error for their results similar to NODE and ANODE papers.

Correctness: Both claims and methodologies are correct.

Clarity: The paper is very well written except few places that have grammatical errors and typos. One example is line 127 in which authors have written: "be formalized in a the following theorem".

Relation to Prior Work: It is clearly discussed how this work is different than the previous ones. Also, authors have compared how their work is different than the previous state-of-the-art results from NODE and ANODE papers.

Reproducibility: Yes

Additional Feedback: Post Rebuttal: Thanks for additional feedback and comments and congrats for the great work.

[Author Response · NeurIPS 2020]

We thank the reviewers for their positive feedback and suggestions. Below we address the major points raised. We will also be sure to incorporate the smaller suggested edits in the final revision.

## Reviewer 1

*Compared with DEQ, the fixed points of MOE are much more stable and unique, which needs experiments.*
Thank you for pointing this out. We will definitely highlight this comparison more in the final version. The original DEQ paper (Bai et al. 2019) uses several training tricks to obtain stable fixed-point convergence, such as gradient clipping, training with subsequences, constraining the weight initialization, and warm start with a pretrained shallow network. There are also anecdotal accounts of difficulty obtaining convergence of DEQs, e.g. Linsley et al. 2020.

In addition, in response to this point we have some additional experiments which we will expand upon further in the final paper. Specifically, we attempted to train a "standard" DEQ with our multi-tier convolutional architecture using Broyden's method and $W$ being unconstrained by the monotone conditions. Using this standard DEQ setup the models would become unstable and diverge after 1-2 epochs, even when using a very large (300) maximum number of Broyden iterations. We will include these results in the paper.

*In Theorem 1, the function f should be convex closed proper (CCP).*
Great point, we mention a bit of discussion around the CCP requirement in Appendix A but will be sure to add it to the theorem statement and discuss in the main paper. Note that all the activations we discuss (e.g. ReLU, tanh, sigmoid, and softplus) can in fact be represented (or approximated) as prox operators of CCP functions.

## Reviewer 2

*NeuralODE and DEQ extend to series data, but it is not clear how to extend MON to such data.*
This is an excellent point. It *is* indeed possible to apply MONs to time-series data by using 1D causal convolutions. There are some subtleties involved in extending MONs to this setting (such as ensuring that influence remains causal when using FFT-based convolutions, which are naively circular), but these can be overcome by careful use of zero padding and other features. We will absolutely highlight this in the updated paper and include some simple experiments.

*In the experimental section, it would be nice to see the results of NODE and ANODE for bigger model size along with the results of MON.*
Absolutely. We ran these experiments over the rebuttal period. Although we hope to run further experiments to verify this behavior, our chief finding so far (which is consistent with discussions we have had about Neural ODEs with others), is that larger ODE models actually begin to *diverge* after a certain point in training. For example, the best performance we could obtain using NODE or ANODE models with ~1M parameters on CIFAR-10 is 72%, which still vastly underperforms MON.

*Could the authors comment on how their approach compares to the "Implicit Deep Learning" paper?*
Great point, and we will add further discussion about this to the paper. The short answer is that the conditions they require more directly relate to stability of the Piccard (simple forward) iteration, and are thus not directly comparable (sometimes the method will be stable while not monotone, and for some monotone operators the naive Picard iteration is not stable, even though other operator splitting methods will be). We will absolutely discuss these points more fully in the paper.

*Is there any intuition on whether such architectures are suitable for regression tasks?*
Yes, since MONs are fundamentally about constructing a fixed point as the hidden representation, we can use a last layer and loss function for regression tasks, just as with normal networks.

## Reviewer 4

*There are still some comparisons that are missing compared to the reported results in NODE and ANODE.*
Hopefully our experiments with larger ODE models (described above) help to address this point. Please let us know if there are additional comparisons we should include in the final version.

*It would be great if authors could report the standard error for their results similar to NODE and ANODE papers.*
Thanks for pointing this out. We have added error bars to the convergence plots for the MON models, which are very narrow around the reported performance. Indeed, this is a major advantage of MON models, which appear relatively stable compared to the (A)NODE models.

[Meta-Review · NeurIPS 2020]

This paper presents an extension to earlier work on deep equilibrium models (DEQ) improving them in several key aspects: using an alternative parameterization that leads to better training properties and is easy to implement, providing a thorough analysis of the proposed approach including convergence guarantees (an aspect that was lacking both theoretically and in practice for DEQ). The reviewers all agreed that this is an important topic in general and that the paper makes a significant contribution - both theoretically and in terms of solid experimental results demonstrating the benefits of the approach. The reviewers did discuss about some additional supporting experiments that would be important to include (comparison to DEQ, NODE/ANODE with larger networks). The authors did provide insight wrt. this in the rebuttal that seems sufficient. As a result the paper should be accepted. I would stress that the additional experiments should make it into the final version of the paper, perhaps by simply extending Table 1 and adding a short note on the problems they encountered experimentally with DEQ.